# The Progress and Trend of Heterogeneous Integration Silicon/III-V Semiconductor Optical Amplifiers

**Wenqi Shi [1,2], Canwen Zou [1], Yulian Cao [1,2,*] and Jianguo Liu [1,2,*]**

1   State Key Laboratory of Integrated Optoelectronics, Institute of Semiconductors, Chinese Academy of Sciences, Beijing 100083, China
2   College of Materials Science and Opto-Electronic Technology, University of Chinese Academy of Sciences, Beijing 100049, China
*   Correspondence: caoyl@semi.ac.cn (Y.C.); jgliu@semi.ac.cn (J.L.)

**Abstract:** Silicon photonics is a revolutionary technology in the integrated photonics field which has experienced rapid development over the past several decades. High-quality III-V semiconductor components on Si platforms have shown their great potential to realize on-chip light-emitting sources for Si photonics with low-cost and high-density integration. In this review, we will focus on semiconductor optical amplifiers (SOAs), which have received considerable interest in diverse photonic applications. SOAs have demonstrated high performance in various on-chip optical applications through different integration technologies on Si substrates. Moreover, SOAs are also considered as promising candidates for future light sources in the wavelength tunable laser, which is one of the key suitable components in coherent optical devices. Understanding the development and trends of heterogeneous integration Silicon/III-V SOA will help researchers to come up with effective strategies to combat the emerging challenges in this family of devices, progressing towards next-generation applications.

**Keywords:** Si photonics; III-V on Si; semiconductor optical amplifiers; heterogeneous integration





## 1. Introduction

With the increase in global data traffic, the electrical interconnections are encountering a huge barrier to satisfying the immense demand for high-speed, low-cost transmission technology in data centers and numerous other emerging applications [1]. In order to overcome the above problems, the idea of using photonic integrated circuits (PICs) to integrate semiconductor laser diodes, optical modulators, amplifiers, multiplexers, waveguides, photodetectors, etc., on a single silicon chip has emerged. Utilizing complementary metal oxide semiconductor (CMOS) manufacturing and packaging technologies, silicon-based devices have the potential for high-volume and low-cost fabrication [2–4]. However, silicon materials have one obvious flaw in optoelectronics attributed to the indirect bandgap structure, as shown in Figure 1 [5]. Hence, the silicon light source is one of the hurdles holding back large-scale optical integration on a silicon platform.

Despite these challenges, scientists have completed a lot of work to obtain light sources on Si substrates over the past few decades. Many approaches to light emission and amplification on silicon substrates have been demonstrated, including Raman lasers [6–8] and Ge-alloy lasers [9–11], showing that group IV optoelectronic devices integrated on Si platforms are possible. However, Raman lasers still require an off-chip light source, and the efficiency of electrically pumping group IV lasers is too low for practical applications compared with group III-V materials [12]. With the persistent efforts of researchers, multiple optical devices have been designed and prepared on silicon substrates through the heterogeneous integration of silicon/III-V. Scientists have realized the functions of light emission, transmission and reception, and finally achieved the photoelectric integration silicon. Combing the best of what silicon and III-V platforms can offer, heterogeneous

integration methods provide many kinds of high-performance passive and active optical devices on a single chip [13–15].

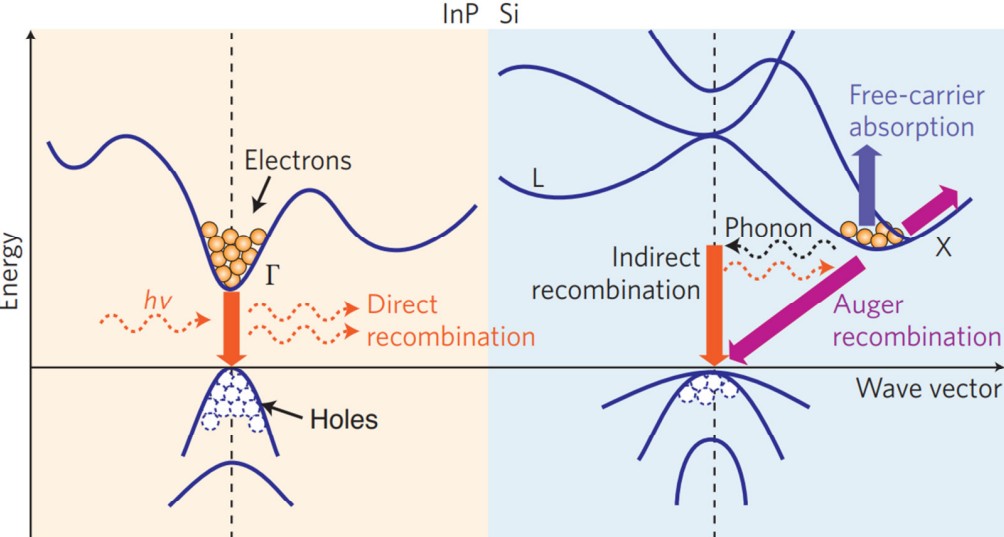

**Figure 1.** Energy band diagrams and major carrier transition processes in InP (**Left**) and silicon crystals (**Right**). Reprinted with permission from Ref. [5]. Copyright 2010 Nature Publishing Group.

Among all of the optical devices integrated on silicon substrates, SOAs are one of the most promising candidates. Silicon-based SOAs with high gain and saturation output power are an essential element in future large-scale silicon PICs. SOAs can compensate for the excess power penalties caused by large numbers of passive components and keep the power of the signal stable, through increasing the output power of each component. Furthermore, SOAs can be used in various optical applications by exploiting their nonlinear properties such as wavelength conversions [16,17] and optical logic design [18,19]. In addition, SOAs can integrate with other devices on the same wafer to improve the performance of the entire photonics system, such as external cavity tunable semiconductor lasers [20–22]. Hence, the research on the heterogeneous integration of SOAs on silicon substrates is of great significance.

In this review, we aim to present the manufacture and applications of heterogeneously integrated SOAs on silicon substrates. We will start by reviewing several approaches to integrating III-V SOAs on silicon in Section 2, including wafer bonding, flip-chip integration, transfer-printing and direct epitaxial growth. Section 3 will review an external cavity tunable semiconductor laser where SOAs are used as light resources or feedback components. In the last section, we draw conclusions regarding the current challenges and provide an outlook for the future development of heterogeneously integrated SOAs on silicon substrates.

## 2. Hybrid Integration of III-V SOAs on Si Substrates

Approaches for realizing the heterogeneous integration of III-V SOAs on Si substrates can be divided into the following four types: flip-chip integration [23,24], wafer bonding [25,26], transfer-printing [27–29] and direct epitaxial growth [30,31]. Flip-chip integration, with the merit of optimizing III-V materials and silicon substrates independently, is considered as the mainstream commercial solution. This method can assemble III-V materials on silicon substrates directly. However, it is not suitable for low-cost manufacturing and dense integration because of its expensive packaging and strict alignment. Benefiting from low-loss evanescent optical coupling, wafer bonding is appropriate for low-cost fabrication. This method transfers III-V material to the silicon-on-insulator (SOI) platform and has the merit of bonding different epitaxial materials onto one single Si substrate [4]. Utilizing the soft poly-dimethyl-siloxane (PDMS) stamps, transfer-printing technology can transfer

the III-V components to silicon substrates. However, this technology is still maturing and is mainly limited to academic research, and so will be difficult to achieve industrial-level production in a short period of time. In addition, transfer-printing technologies also face problems including transfer yield, large-scale transfer and alignment issues, which could result in poor bonding quality. Direct epitaxial growth technology can grow III-V quantum-dot (QD) gain materials on silicon substrates. The lattice constant mismatch between III-V materials and silicon has largely been reduced through the development of material growth technologies. Furthermore, SOAs manufactured through this method exhibit excellent performance including high saturation output power, high-temperature stability and fast gain response. However, this technology still has a long way to go to achieve industrial-level production. The critical parameters of hybrid integration SOAs are shown in Table 1.

**Table 1.** Brief survey of silicon-based SOAs.

| Integration Technology | Unsat. Gain (dB) | Output Power (dBm) | Current (mA) | WPE (%) | Length (mm) | References | Peak Gain Wavelength (nm) |
|---|---|---|---|---|---|---|---|
| Wafer bonding | 13 | 11 | 200 | 5.25 | 1.36 | [32] | 1575 |
| Wafer bonding | - | 11 | - | 12.1 | 0.4 | [33] | 1540 |
| Wafer bonding | 20 | 17 | 100 | - | 0.7 | [26] | 1284 |
| Wafer bonding | 27 | 17.24 | 300 | - | 1.45 | [25] | 1575 |
| Wafer bonding | 10 | 13 | 110 | - | - | [34] | 1549 |
| Flip-chip | 10 | 10 | 150 | - | 0.75 | [35] | 1550 |
| Flip-chip | 23 | - | 100 | - | 0.8 | [24] | 1550 |
| Transfer-printing | 23 | 9.6 | 140 | - | 1.35 | [27] | 1548 |
| Transfer-printing | 17 | 11.8 | 160 | - | 1.35 | [27] | 1548 |
| Transfer-printing | 14 | 9 | - | - | - | [28] | 1570 |
| Direct epitaxial growth | 34.1 | 24.1 | - | 19.7 | - | [36] | 1320 |
| Direct epitaxial growth | 39 | 23 | 750 | 14.8 | 5 | [36] | 1315 |

## 2.1. Wafer Bonding Technology

The first hybrid III-V/Si SOA formulated through wafer bonding was reported by Park from UCSB [32]. Most of the optical mode is confined to the Si waveguide and transferred to the gain region which consists of a quantum well (QW) through evanescent coupling, as shown in Figure 2a. Furthermore, in order to decrease the reflection, they designed the Si waveguide at an angle of 7° from the normal of the output facet. The hybrid SOA has a maximum gain of 13 dB, as well as a saturation output power of 11 dBm. Figure 2b shows the cross-sectional image of the fabricated SOA.

Manufacturing III-V/Si SOAs through the oxygen plasma-assisted method is a typical approach. The entire process flow is shown in Figure 2c. In order to obtain clean surfaces, HF and NH$_4$OH solutions are utilized to remove the impurity in III-V and Si materials. The bonded sample is annealed at a temperature of 300 °C and a pressure of 1.5 MPa for 12–18 h to achieve strong covalent bonds. This process can be divided into Equations (1) and (2).

$$Si - OH + M - OH \rightarrow Si - O - M + H_2O(g) \qquad (1)$$

$$Si + 2H_2O \rightarrow SiO_2 + 2H_2(g) \qquad (2)$$

where *M* represents a high-electronegativity metal (such as group III and IV).

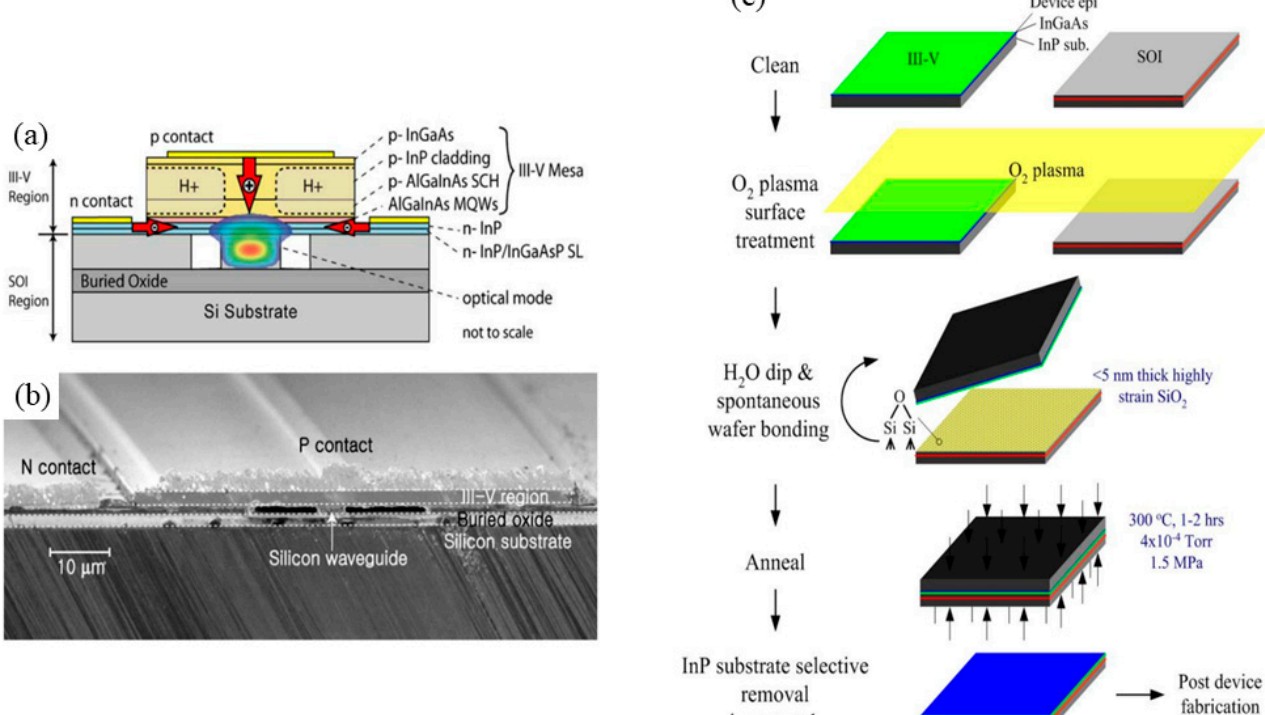

**Figure 2.** (**a**) Wafer bonding structure cross section; (**b**) SEM image. Adapted from an open access source, Ref. [37]; (**c**) schematic of the $O_2$ plasma-assisted low-temperature III-V to Si bonding process flow. Reprinted with permission from Ref. [38]. Copyright 2010 WILEY-VCH Verlag GmbH & Co. KGaA, Weinheim, Germany.

The effective use of divinylsiloxane-bis-benzocyclobutene (DVS-BCB) bonding to realize hybrid SOAs was reported by Ghent University [39], as shown in Figure 3a. In order to reduce the power consumption, they designed a structure with high confinement in the III-V region. Furthermore, they optimized the thickness of the bonding layer down to 40 nm to reduce the influence caused by heat accumulations. Their device realized an on-chip gain of 13 dB with a drive current of 40 mA at room temperature. In order to obtain a higher confinement factor in the active region, Kaspar et al. [40] reported a hybrid III-V/Si on-chip SOA fabricated through BCB bonding in 2014. The optical mode evanescently coupled III-V and Si through a taper structure. They designed the Si waveguide at an angle of 10° from the normal of the output facet. Their device demonstrated 10 dB maximum fiber-to-fiber gain and 28 ± 2 dB maximum internal gain.

Decreasing the optical coupling loss is also an effective way to improve the performance of SOAs. Cheung et al. [33] from UCD reported a hybrid SOA with high wall-plug efficiency (WPG), as shown in Figure 3b. In order to make sure the bonding surface clean, they designed vertical outgassing channels (VOCs) to remove byproducts generated in the bonding processes. They experimentally obtained a 400 μm-long flared SOA which provided 12.1% wall-plug efficiency (WPE) with output power > 10 mW, as well as a 400 μm-long straight SOA which provided 7.3% WPE with output power < 10 mW.

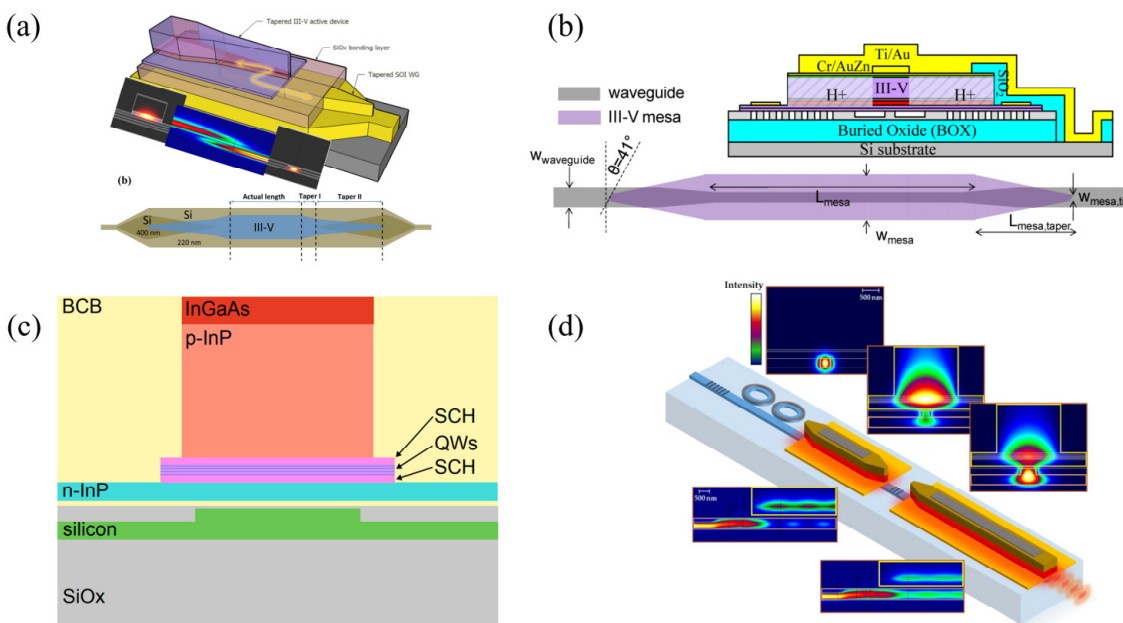

**Figure 3.** (**a**) Three-dimensional view and schematics of the hybrid SOA designed by Keyvaninia from UCSB. Adapted from an open access source, Ref. [41]; (**b**) cross-section of the SOA waveguide designed by Cheung. Adapted from an open access source, Ref. [33]; (**c**) cross section and top view of SOA designed by Kasper. Adapted from an open access source, Ref. [25]; (**d**) 3D schematic view of the III-V/Si integrated tunable laser-SOA. Adapted from an open access source, Ref. [34].

Kasper et al. [25] from Ghent University reduced the confinement in the active region to manufacture a hybrid SOA with high saturation output power, as shown in Figure 3c. In order to decrease the interaction between the side wall and the guide mode, they designed the III-V gain region to be 0.5 μm wider than the p-InP mesa. Their devices demonstrated 27 dB small signal gain and 17.24 dBm saturation power. In 2021, Ramirez et al. [34] from Paraiso III-V Laboratory reported the co-integration of the III-V/Si laser-SOA through wafer bonding technology, as shown in Figure 3d. The compact SOA demonstrated 13 dBm on-chip output power and 10 dB net gain with an injected current of 110 mA.

Benefiting from the low-loss evanescent optical coupling, wafer bonding technology can realize easier and low-cost mass manufacturing. However, for direct bonding, the creation of byproducts during the wafer bonding process is still an intractable problem which affects the performance of optical devices. It is important to overcome this problem to obtain a high-quality bonding layer through optimizing the size and distribution of VOCs. For indirect bonding such as DVS-BCB bonding, due to the low thermal conductivity of the bonding layer, heat accumulations are difficult to dissipate. One possible solution is connecting the bonding layer with the heat-sink structure through metal contacts. In addition, reducing the thickness of bonding layer to sub-100 nm is also an effective strategy to overcome substantial heat accumulation.

### 2.2. Transfer-Printing Technology

The first controlled and massively parallel transfer of micron-scale materials from source wafers to target substrates was reported by Rogers and his co-workers from the University of Illinois in 2004 [42,43]. This technology uses PDMS rubber as a carrier to transfer marked molds. PDMS rubber is soft and elastic, and it can handle fragile materials without damage, as shown in Figure 4. By changing the adhesion force between the transfer target and the carrier through controlling the temperature, the substrate and the transfer target could achieve the purpose of picking and printing [44].

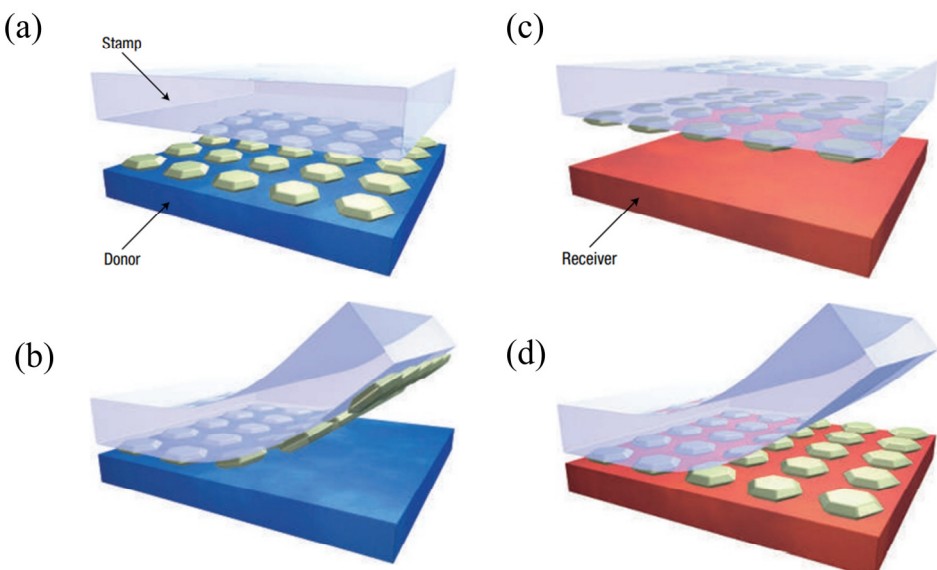

**Figure 4.** Overview of the transfer print process: (**a**) prepare donor substrate and apply rubber stamp; (**b**) quickly peel back stamp and grab objects from the donor; (**c**) apply inked stamp to receiving substrate; (**d**) slowly peel back stamp and print objects onto the receiver. Reprinted with permission from Ref. [43]. Copyright © 2005, Nature Publishing Group.

The large refractive index differences between III-V materials and silicon have caused problems for the integration of hybrid SOAs on a silicon chip. Beeck et al. [28,45] from Ghent University reported an approach to solve this refractive index mismatch problem through using an intermediate layer of hydrogenated amorphous silicon, as shown in Figure 5. First, they used low-pressure chemical vapor deposition (LPCVD) to deposit $SiO_2$ on silicon substrates. Subsequently, they employed plasma-enhanced chemical vapor deposition (PECVD) technology to deposit a thin $SiO_2$ layer and hydrogenated amorphous silicon on $Si_3N_4$ platforms. The position of the micro-transfer printing is defined in the hydrogenated amorphous silicon layer. In this process, the $Si_3N_4$ layer is protected by the $SiO_2$ layer in the etching process and then the $Si_3N_4$ layer is patterned by electron beam lithography (EBL) technology. Finally, the III-V gain region can be integrated on waveguides through transfer printing technologies. Their hybrid SOA devices demonstrated 8 mW saturation power and 14 dB gain with the input current of 120 mA. Furthermore, their approach may be suitable for other low-refractive-index platforms such as lithium niobate.

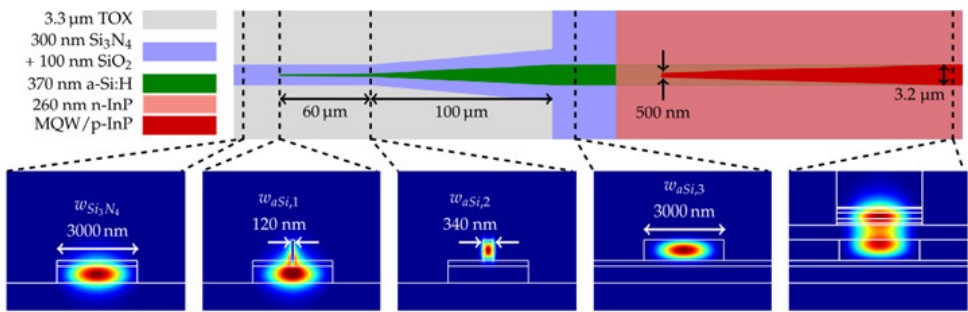

**Figure 5.** Schematic of the taper from the $Si_3N_4$ waveguide to the III-V region designed by Beeck. Adapted from an open access source, Ref. [28].

In 2020, Haq et al. [27] reported the preparation of dense arrays of hybrid SOAs fabricated by transfer-printing, as shown in Figure 6. The length of the SOAs was 1.35 mm and the width was 40 μm. Dense III-V SOA arrays were manufactured on InP wafers, which could be transfer-printed on the target SOI photonic circuit in a massively parallel

manner. Furthermore, this method had great flexibility in integrating different epitaxial layers onto the same silicon platform without changing the casting process flow. Their devices demonstrated a small signal gain of 23 dB and a saturation power of 9.2 mW with 140 mA input current and 17 dB small signal gain and 15 mW saturation power with 160 mA input current.

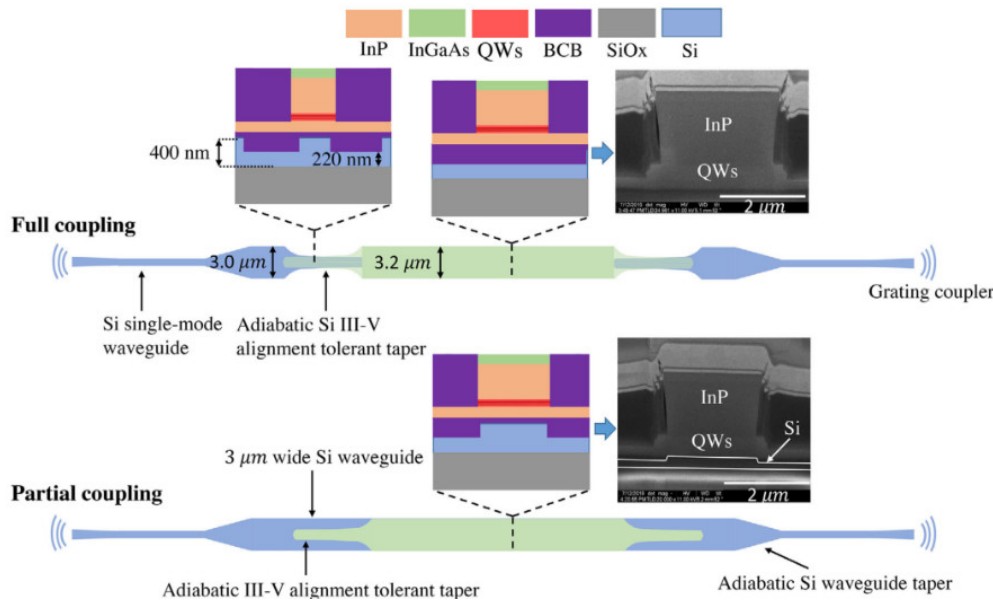

**Figure 6.** Schematics and cross sections of the hybrid III-V/Si SOAs designed by Haq. Adapted from an open access source, Ref. [27].

Transfer-printing technology can integrate different III-V devices on a single silicon substrate. Furthermore, it can realize high density integration and direct integration of waveguide-in/waveguide-out devices such as SOA. This technology for the hybrid integration of III-V/silicon photonics devices can enable the creation of more complex and powerful chip-scale photonic systems.

*2.3. Flip-Chip Integration Technology*

Flip-chip technology is defined as a chip attached to the pads of a substrate or another chip with various interconnecting materials and methods [46], as shown in Figure 7. This technology was introduced in the early 1960s. Presently, its applications have been extended to face-to-back, chip-to-chip and face-to-face [47]. Owing to the better integration flexibility and better heat conduction between III-V materials and silicon substrates, flip-chip integration is more suitable for applications at high temperatures. Using these advantages, Tanaka et al. [23] from Fujitsu developed an SOA with high output power and WPE through flip-chip integration technology, as shown in Figure 8a. In order to achieve low-loss optical coupling, it is of vital importance to match the mode field between the SOA and the waveguides. They compensated for the influence of the welding position by introducing a constant welding offset. The distribution of horizontal misalignment is plotted in Figure 8b. They optimized the height relationship between the SOA and waveguides by controlling the thickness of each layer on chips. The vertical misalignment results are shown in Figure 8c. The alignment error is ±0.9 μm, and the measured value is within 1 dB. Finally, their SOA devices demonstrated 11.7 dBm output power with a threshold current of 9.4 mA and a WPE of 7.6 at 20 °C, as shown in Figure 8d. Furthermore, benefiting from the high heat conductance, the output power can maintain over 10 dBm at 60 °C.

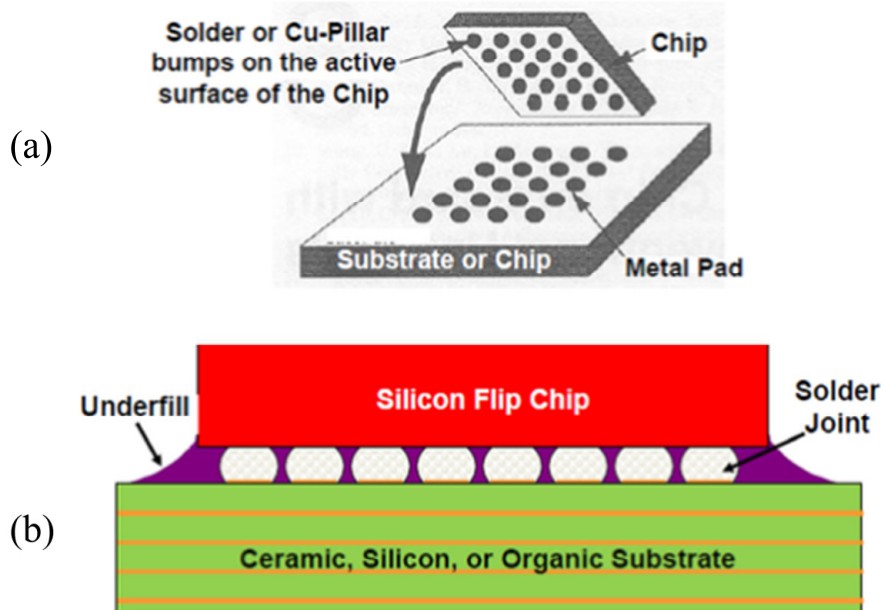

**Figure 7.** (**a**) Definition of flip-chip assembly; (**b**) flip-chip assembly on various substrates. Reprinted with permission from Ref. [46]. Copyright © 2018, Springer Nature Singapore Pte Ltd., Singapore.

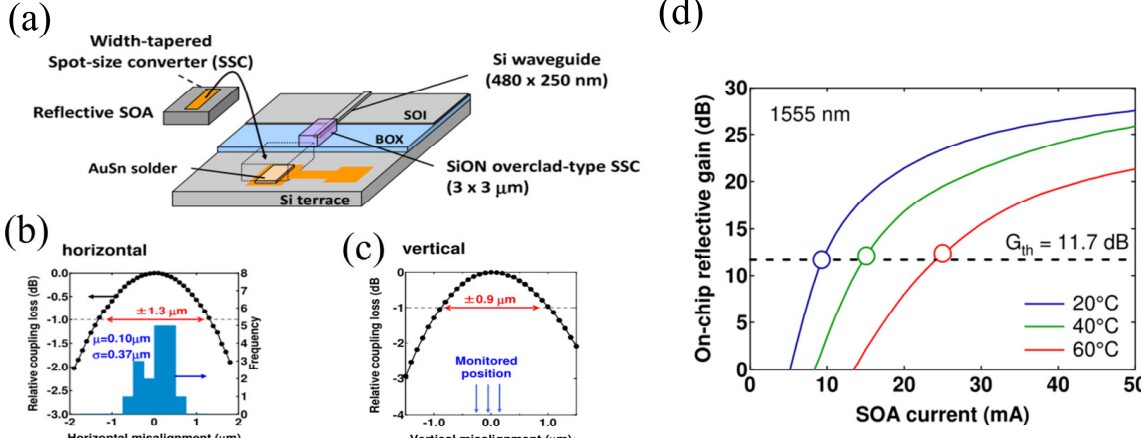

**Figure 8.** (**a**) Schematic of structure of the SOA interface; (**b**) distribution of horizontal misalignment and coupling tolerance; (**c**) difference in waveguide heights and vertical coupling tolerance; (**d**) on-chip gain current characteristics of reflective SOA. Adapted from an open access source, Ref. [23].

By utilizing passive alignment through the marks on both chips, Matsumoto et al. [24] manufactured hybrid III-V/silicon SOAs with small coupling loss. They realized a gain of 10 dB with the transmission loss of 0.2 dB, and the coupling loss was less than 3 dB. After the technological improvement and new array design, they reported the demonstration of a lossless Si switch which was realized by an SOA array through flip-chip integration within than less than ±1 μm alignment accuracy. The SOA exhibited a linear gain of 15 dB and a coupling loss of 7.7 dB, including a 5.1 dB alignment loss and a 2.6 dB coupling loss between the Si waveguide and III-V region [24]. Figure 9a is a schematic diagram of the hybrid integrated InP-SOA matrix switch; for inline amplification, both input and output waveguides of these SOAs are coupled to Si waveguides, as shown in Figure 9b. An isotropic mode field with a diameter of about 3 μm was obtained at the interface between the SOA and optical platform. Figure 9c shows the relationship between the calculated mode field diameter and the tapered tip width, and the test result is shown in Figure 9d.

Further optimization of the waveguide groove depth to reduce the loss in the vertical direction is expected to increase the maximum gain of the SOA chip to 23 dB.

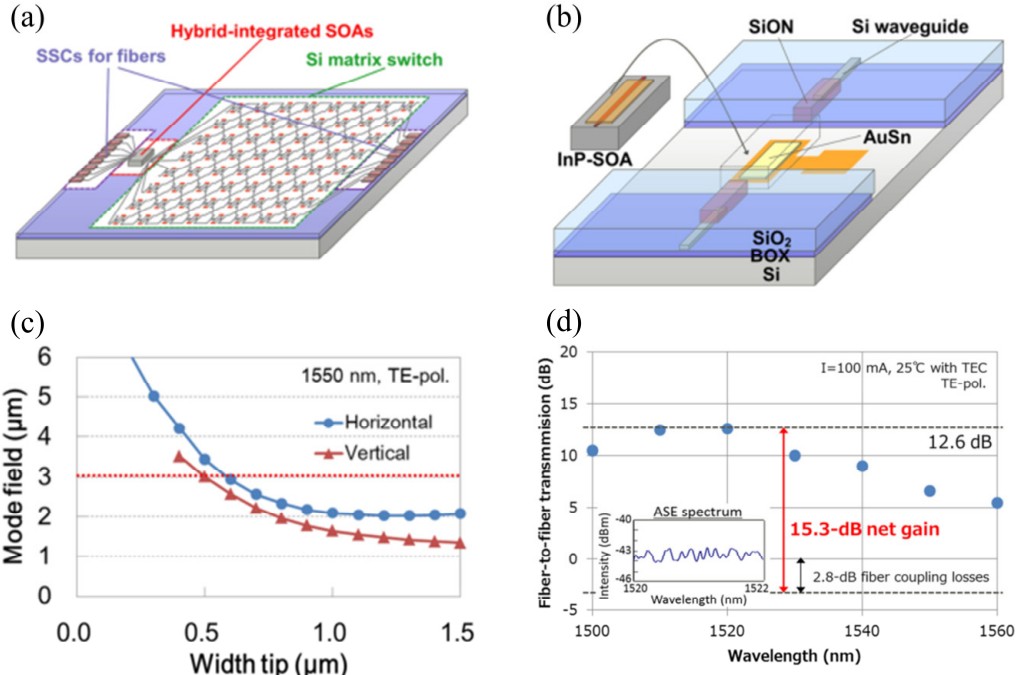

**Figure 9.** (**a**) Schematics of a Si switch with hybrid SOAs and (**b**) configuration of flip-chip technology for amplification; (**c**) calculated mode field diameter of SOAs; (**d**) transmission spectrum of the chip. Inset shows tested ASE spectrum around 1521 nm. Adapted from an open access source, Ref. [24].

To date, flip-chip technology is the current mainstream commercial solution of realizing on-chip SOA integration, and it has the advantages of high light-emitting efficiency and quality. Although scientists have improved the alignment accuracy effectively and have obtained optical devices with high performance, the high cost of alignment to couple the light leads to this technology not being not suitable for low-cost mass manufacturing and dense integration. Optimizing the integration technology to reduce the alignment difficulty or exploring novel designs to decrease the influence of the misalignment could make it possible to mitigate the above problems of flip-chip integration technology.

### 2.4. Direct Epitaxial Growth Technology

Growing single-crystal thin films on crystal-oriented wafers has become a critical technique to improve modern photonic devices on a variety of inorganic substrates [48–50]. Epitaxy is a process by which a thin layer of one crystal is deposited in an ordered fashion onto a substrate crystal. Generally, homogeneous epitaxial growth manufactures single-crystal epitaxial layers with a high quality through replicating the crystal structure of the substrate. In contrast, heteroepitaxial epitaxial growth is usually limited by the lattice mismatch between the substrate and the epitaxial layer [51]. Hence, III-V/Si hybrid integration would suffer from a high density of defects [52].

In order to reduce the influence of the above problems, scientists have developed various epitaxial growth methods such as domain-matched epitaxy [53–55] and epitaxial lateral overgrowth [56–58] and introduced several new types of buffer layers such as low-temperature buffer layer [59–61], lattice-engineered buffer layer [62,63], and metamorphic buffer layer [64,65]. These novel methods allow a wide variety of compound semiconductors to grow on lattice mismatched substrates. However, owing to unavoidable buffer layers and extra optical loss generated in the mismatching areas, it is difficult to achieve highly efficient light coupling between the III/V and silicon regions [4]. With the increasing

demand for heterogeneous material integration, various techniques have been exploited, including epitaxial lift-off (ELO), mechanical peeling, laser peeling, and two-dimensional material auxiliary layer transfer (2DLT), as shown in Figure 10.

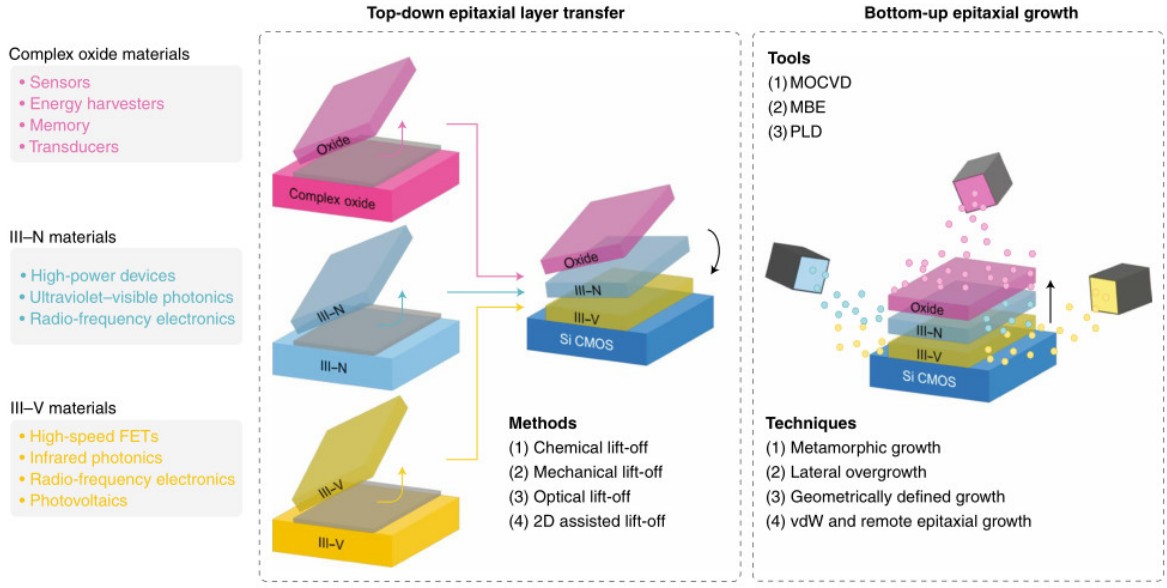

**Figure 10.** Overview of heterogeneous epitaxial integration of mismatched materials for photonic devices. Reprinted with permission from Ref. [52]. Copyright 2019 American Chemical Society. Copyright 2019, the author(s), under exclusive license to Springer Nature Limited, Berlin, Germany.

Owing to their unique characteristics, QDs are considered as promising candidates for creating high-performance optical devices [66,67]. QDs as gain material in SOAs have many advantages, such as a low confinement factor, low internal loss, low threshold current density and fast carrier dynamics [68]. The fast gain response makes it suitable to amplify high-speed signals without pattern effects [69]; the low threshold current density, internal loss, and confinement factor contribute to low-noise-figure operation [70,71].

In 2019, Liu et al. [36] from the University of California reported a hybrid integrated SOA working in O-band on a silicon substrate using the direct epitaxial growth technique for the first time, as shown in Figure 11a,b. They employed a tapered gain region design to enhance the saturation output power.

Figure 11c shows the relationship between the on-chip input power and gain at different temperatures. The peak gain of 39 dB was obtained at 20 °C, and the peak gain values at 70 °C, 60 °C and 40 °C were 23.4, 25.8 and 34.1 dB, respectively. The relationship between on-chip output power and input power is shown in Figure 11d. It can be seen that their device can provide >20 dBm output power under all temperature conditions with the input power is 0 dBm. A 19.7% WPE value could be obtained with a 2.3 mW input power at 40 °C. The NF values varied from 6.6 to 9.1 dB in the low-input-power range for all conditions. Their work has a wide range of potential applications in high-gain, high-output power and high-temperature devices.

Yan et al. [72] from Cornell reported the growth and integration of niobium nitride (NbN)-based superconductors with the wide-bandgap family of semiconductors (SiC, GaN and AlGaN) through molecular beam epitaxy (MBE) technology. They observed in the transistor's output characteristics a negative differential resistance, which could be used in amplifiers. Their work provides a novel direction for the manufacture of hybrid III-V/Si on-chip SOAs.

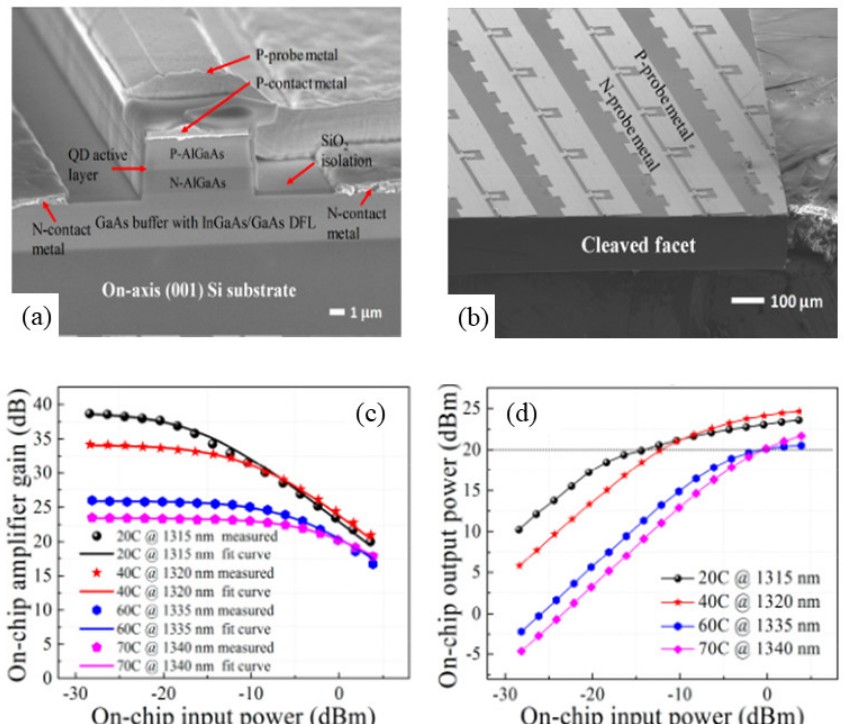

**Figure 11.** (**a**) Cross-sectional views of the QD-SOA and (**b**) a corner of the fabricated SOA array. Copyright 2019, American Chemical Society; (**c**) the relationship between the on-chip gain and input power at 20, 40, 60, and 70 °C; (**d**) the relationship between the on-chip output power and input power at 20, 40, 60, and 70 °C. Reprinted with permission from Ref. [36]. Copyright © 2019, American Chemical Society.

## 3. The Application of Silicon-Based SOAs

Photonic integration brings a promise of significant cost, power and space advantages in present optical data transmission applications. Over the past decade, effective and low-cost silicon-based wafers have been considered as promising candidates to accommodate multiplicate optically functional materials such as Group IV and III-V materials [1]. Silicon photonics has the advantages of low propagation loss and high integration density. On the other hand, III-V materials with high gain values can be flexibly used in bandgap engineering by changing the composition to realize high-performance light sources [73]. The performance of optical devices is further improved by implementing a variety of devices on a single chip using heterogeneous technology, where all optical connections are on-chip without external alignment. In this section, we will introduce an external cavity tunable semiconductor laser where an SOA is used as a light resource or a feedback component.

Wavelength tunable semiconductor lasers have a wide range of important applications, such as fiber-optic communications [74,75], optical sensing [61,76] and atomic clocks [77–79]. They are also considered as an optical power/signal supplying source, which is one of the most fundamental elements in optical wavelength division multiplexing (WDM) communication systems [80–83]. Under certain conditions, optical feedback can enhance, prolong or suppress the relaxation oscillation in the transient output, and improve the linewidth quality of the laser.

In 2015, using an SOA as the light source, Srinivasan et al. [21] from the University of California designed a silicon-based tunable laser with micro-rings as the feedback component, as shown in Figure 12a. In order to realize the function of tuning wavelength, they designed individual heaters on each ring to control the feedback resonance wavelength. Furthermore, they also optimized the size of rings to avoid thermal cross talk. Their devices demonstrated 160 kHz linewidth and over 11.8 dBm output power. The side mode suppression ratio (SMSR) was over 40 dB in the whole tuning range of 29 nm.

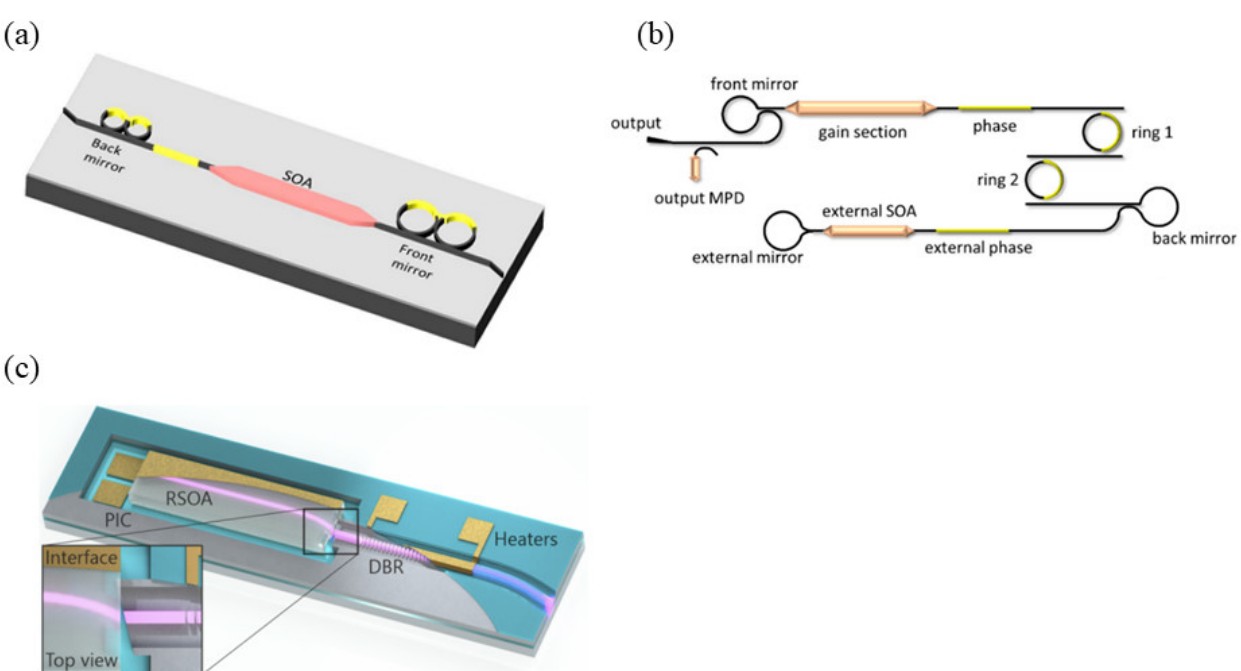

**Figure 12.** Structure of wavelength tunable laser designed: (**a**) by Srinivasan. Adapted from an open access source, Ref. [21]; (**b**) by Komljenovic. Adapted from an open access source, Ref. [84]; (**c**) by Zia. Adapted from an open access source, Ref. [22].

In the same year, Komljenovic et al. [20] from the University of California realized an tunable laser on low-loss silicon waveguide platforms with micro-ring resonators as the feedback component, as shown in Figure 12b. The gain section is the light source and SOA provides the external cavity feedback. Their result demonstrated tuning in excess of 54 nm in the O-band and a significant reduction in linewidth in the laser owing to the feedback from the external cavity. The linewidth is <100 kHz in the whole tuning range and the minimal value is around 50 kHz. Recently, Zia et al. [22] from Tampere University reported the first flip-chip integration of a GaSb SOA on silicon photonic circuits. The integrated hybrid laser is shown in Figure 12c, and it comprises a reflective SOA (RSOA) coupled with a DBR grating which can be tuned to the feedback through heaters. In particular, they exploited 3 µm-thick SOI waveguide technology which has the advantage of easily achieving low coupling loss between GaSb and silicon waveguides without the use of spot size converters (SSCs). The on-chip hybrid laser demonstrated an output power of 7.8 dBm at room temperature. The peak wavelength could be tuned between 1983 nm and 1990 nm and the SMSR changed from 32 dB to 37 dB in the whole tuning range. Their result opens up attractive prospects for the development of PICs in a broad spectral range extending to 3 µm.

SOAs can also be used in various optical applications by exploiting their nonlinear properties such as wavelength conversion. A silicon-based SOA-Mach–Zehnder interferometer (MZI) wavelength converter was reported by Mitsubishi Electric Corporation in 2015 [85]. With the feedback control, they demonstrated a dynamic range of >8 dB for TE and TM polarized input signals non-return-to-zero (NRZ) modulated at 43 Gb/s. In 2016, Zhejiang University reported 12.5 Gb/s all-optical wavelength conversion (AOWC) for wavelength up- and down conversion based on hybrid III-V/silicon SOAs. Furthermore, their results demonstrated that with the power consumption of the SOA being <250 mW, the converter could realize 6 pJ/bit energy consumption at 40 Gb/s. The hybrid integration method involves DVS-BCB bonding [86], as shown in Figure 13.

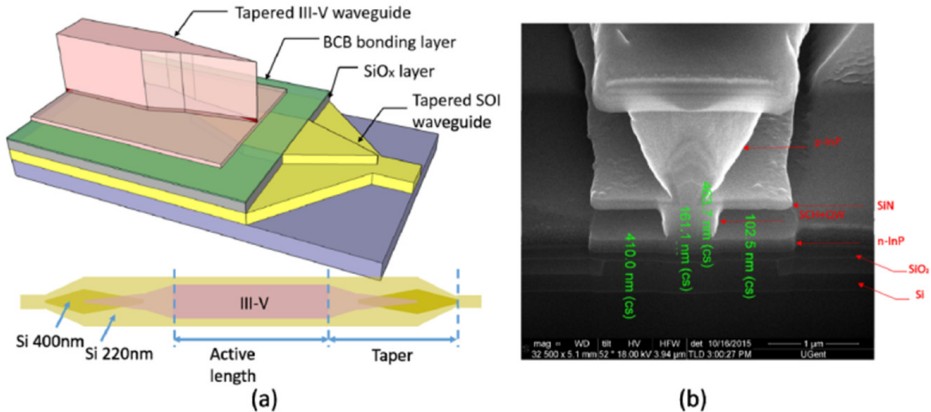

**Figure 13.** (**a**) Schematics and top views of the SOA; (**b**) cross section of the taper tip. Adapted from an open access source, Ref. [86].

## 4. Conclusions and Outlook

In the 1950s, the invention of lasers triggered a technological revolution. III-V semiconductors were subsequently implemented in diode lasers, and scientists began research on SOAs a few years later. SOAs can compensate for the excess power penalties caused by large numbers of passive components and keep the power of the signal stable, through increasing the output power of each component. Silicon photonics has improved rapidly over the past three decades, owing to the need for more complex, higher-function, and lower-cost photonics-integrated circuits. Although several SOAs have been developed on InP platforms and hybrid III-V/silicon-based platforms, achieving high gain and high output power is still a major challenge.

In terms of the previous work, we reviewed four types of technologies applied in hybrid integration SOA, including wafer bonding, flip-chip integration, micro-transfer printing and direct epitaxial growth. These technologies have been extensively explored and optimized according to the needs of different optical devices. Flip-chip technology is the current mainstream commercial solution, but the precise alignment to couple the light can lead to high packaging costs, and so it is not suitable for low-cost mass manufacturing. Benefiting from low-loss evanescent optical coupling, wafer bonding technologies are suitable for low-cost manufacturing, but the byproducts produced in the bonding process still represent an intractable challenge. Transfer-printing and direct epitaxial growth technologies have been demonstrated to have good performance in heterogeneous integration, especially QD SOAs integrated through the latter method, which have shown a gain of up to 39 dB. However, these two technologies still have a long way to go to achieve industrialization. Hence, it is of vital importance for scientists to explore new heterogeneous integration approaches and design new structures. For example, for wafer bonding technology, the byproducts generated in the bonding process affect the surface roughness. Therefore, it is important to optimize the size and distribution of VOCs to achieve a high-quality bonding layer. Furthermore, the heat dissipation is another intractable problem, especially in DVS-BCB bondings. One possible solution is connecting the bonding layer with the heat-sink structure through metal contacts. In addition, reducing the thickness of the bonding layer to sub-100 nm is also an effective strategy to overcome substantial heat accumulation.

SOAs can be integrated with other devices on the same wafer to improve the performance of the whole photonics system, such as external cavity tunable semiconductor lasers, which also represent an important research direction of on-chip SOAs. Therefore, SOAs are considered as promising candidates for future light sources in coherent optical devices. There is still room for on-chip SOAs to improve the properties of silicon-integrated optical circuits by modifying the heterogeneous integration technology. The next generation of SOAs may realize high-density integration with high net gain, low coupling loss and low noise figures.

**Author Contributions:** W.S. wrote the manuscript, C.Z. created the figures, Y.C. and J.L. conceived, designed and edited the manuscript. All authors have read and agreed to the published version of the manuscript.

**Funding:** This work was supported by the National Key R&D Program of China (No. 2021YFB2800402).

**Institutional Review Board Statement:** Not applicable.

**Informed Consent Statement:** Not applicable.

**Data Availability Statement:** Not applicable.

**Acknowledgments:** The authors would like to acknowledge Yejin Zhang for their insightful discussions.

**Conflicts of Interest:** The authors declare no conflict of interest.

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
