# Peer review of "The Progress and Trend of Heterogeneous Integration Silicon/III-V Semiconductor Optical Amplifiers"

_photonics, doi:10.3390/photonics10020161_

Round 1

Reviewer 1 Report

I believe that the authors have written a very interesting and timely review on the topic of semiconductor optical amplifiers based on the new silicon-based platforms. Even for people who do not work in this field of photonics, the structure of the paper allows to understand all the main directions of SOA development, technological methods, as well as existing and potential applications of semiconductor optical amplifiers. At the same time, all technological methods are well illustrated and clearly stated. After reading this review, there is a feeling that this is really a very important and rapidly developing area of exploration of more sophisticated heterogeneous integration technologies. I think the paper can be published in the present form.

Author Response

Dear reviewer,

Thank you for your careful reading as well as the thoughtful and constructive comments earnestly. We have revised some grammatical and spelling mistakes in the manuscript.

Sincerely,

Wenqi Shi

Reviewer 2 Report

This manuscript thoroughly reviewed the progress and the trend of current technologies for Si/III-V SOAs, which is of interest to the readers of Photonics.  there I recommend it to be published.  One suggestion is to proofread the manuscript one more time to fix some grammar mistakes.

Author Response

(The authors gave the same response as above.)

Reviewer 3 Report

The authors reviewed recent improvement in both manufacture and application of integrated semiconductor optical amplifiers (SOAs) on Si. Four major manufacture technologies with different optimizing aims, such as wafer bonding, flip-chip integration, micro-transfer printing, and direct epitaxial growth, are well discussed and summarized. However, the review of the application of SOAs is limited to tunable lasers, and thus lacks generality required for a review paper. Please find my specific comments/questions below:

1.     For Section 2, the authors provided discussion based on citation of two journal papers for each mentioned technology. It is recommended to add intuitive and integrated figures and more literatures for better presentation.

2.     The application of SOAs should not be limited to tunable lasers. It is recommended to discuss more in application improvement of SOAs, such as wavelength conversion, logic gates, etc.

3.     The trend on heterogeneous integration of III-V SOAs on Si substrates can be explored in depth, especially the challenges with the four mentioned integration methods, because the content on this aspect is too short. For example, the authors mentioned that we need to design new structures for heterogeneous integration, where the authors can point out detailed valuable solutions or possible development directions.

4.     Detailed comparison among the four methods can be provided in a table with key parts clearly listed.

5.     Can the authors address the coupling efficiency between III-V SOAs on Si substrates and other optical components with the four methods, their advantages in this regard, and further development directions?

6.     Chapter 3 has the title of “the application of semiconductor optical amplifiers in tunable lasers”. Such a title is quite confusing because it seems that it reviews a different type of tunable lasers from on-chip lasers. Is it true? Because the structures in Fig. 15 is a common structure for on-chip lasers. Besides, in the structures described in Figs. 12 and 13, SOA and external resonators are not on the same chip. Is it appropriate to name it as monolithic integration?

7.     In Chapter 2.4, the authors claimed, “Due to the characteristics of zero-dimensional carriers of quantum dots, the fabrication of SOAs in CMOS is barely affected by the dislocation result from lattice constant mismatch between III-V and Si materials.” However, the reviewed paper cannot support the claim. In Fig. 12, the QD lies between N-AlGaAs and P-AlGaAs.

8.     Why did the authors claim that wafer bonding does not need a strict alignment? The alignment between III-V materials and Si materials should also influence the SOA performance.

Author Response

Dear reviewer,

Thank you for your careful reading as well as the thoughtful and constructive comments earnestly, which certainly help us to improve the quality of our manuscript. All comments are addressed on a point-point basis below, as shown in the attachment.

Sincerely thank you very much for your comments and suggestions, we have benefited a lot from your comments and we hope meet your approval.

Sincerely,

Wenqi Shi

.
